# Numerical Simulation of Polyacrylamide Hydrogel Prepared via Thermally Initiated Frontal Polymerization

**DOI:** 10.3390/polym16070873

**Published:** 2024-03-22

**Authors:** Xiong Yi, Shengfang Li, Pin Wen, Shilin Yan

**Affiliations:** 1Hubei Key Laboratory of Theory and Application of Advanced Materials Mechanics, Wuhan University of Technology, Wuhan 430070, China; ykxalmj@163.com (X.Y.);; 2School of Chemistry and Chemical Engineering, Hubei Polytechnic University, Huangshi 435003, China

**Keywords:** frontal polymerization, acrylamide hydrogel, curing kinetics, numerical simulation, DSC

## Abstract

Traditional polymer curing techniques present challenges such as a slow processing speed, high energy consumption, and considerable initial investment. Frontal polymerization (FP), a novel approach, transforms monomers into fully cured polymers through a self-sustaining exothermic reaction, which enhances speed, efficiency, and safety. This study focuses on acrylamide hydrogels, synthesized via FP, which hold significant potential for biomedical applications and 3D printing. Heat conduction is critical in FP, particularly due to its influence on the temperature distribution and reaction rate mechanisms, which affect the final properties of polymers. Therefore, a comprehensive analysis of heat conduction and chemical reactions during FP is presented through the establishment of mathematical models and numerical methods. Existing research on FP hydrogel synthesis primarily explores chemical modifications, with limited studies on numerical modeling. By utilizing Differential Scanning Calorimetry (DSC) data on the curing kinetics of polymerizable deep eutectic solvents (DES), this paper employs Malek’s model selection method to establish an autocatalytic reaction model for FP synthesis. In addition, the finite element method is used to solve the reaction–diffusion model, examining the temperature evolution and curing degree during synthesis. The results affirm the nth-order autocatalytic model’s accuracy in studying acrylamide monomer curing kinetics. Additionally, factors such as trigger temperature and solution initial temperature were found to influence the FP reaction’s frontal propagation speed. The model’s predictions on acrylamide hydrogel synthesis align with experimental data, filling the gap in numerical modeling for hydrogel FP synthesis and offering insights for future research on numerical models and temperature control in the FP synthesis of high-performance hydrogels.

## 1. Introduction

Thermoset polymer-based composite materials are used in various engineering applications due to their excellent mechanical properties and environmental resistance. However, composite material manufacturing processes based on autoclaves and heated molds require significant capital investment, and large-scale curing processes involving high temperature and pressure necessitate complex, time-consuming curing cycles and substantial energy expenditure [1]. Frontal polymerization (FP) represents a valuable alternative synthesis approach for polymer-based composites. This reaction features a highly localized and self-propagating exothermic reaction zone, representing a cheaper, faster, and more energy-efficient option for composite material manufacturing [2]. It enables the rapid fabrication of certain composite material components, efficiently overcoming the limitations of traditional high-temperature, high-pressure curing processes [3].

Numerous types of polymers can undergo frontal polymerization. The primary characteristic of FP is that it is highly exothermic, and the reaction rate must be sufficiently high so that the energy released far exceeds the thermal energy lost to the environment. The FP reactions explored to date mainly include the anionic and cationic polymerization of epoxy resins [4,5,6], free radical polymerization of olefins [7,8,9], addition polymerization of polyurethanes [10,11,12], and ring-opening metathesis polymerization of dicyclopentadiene (DCPD) [13,14,15]. Acrylamide hydrogels synthesized based on front-end polymerization have a promising background in biomedical applications and prospects for 3D printing, and they have already been widely applied in various fields [16]. Jiang Yang et al. [17,18] have used acrylamide (AAc), acrylamide (AM), and choline chloride (ChCl) in specific molar ratios to synthesize deep eutectic solvents (DES) for the preparation of macroporous polyacrylamide hydrogels. This method surpasses conventional techniques, emphasizing simplicity, environmental sustainability, rapidity, and energy conservation. At ambient temperature, introducing an initiator and cross-linker into DES rapidly solidifies monomer solutions into hydrogels, demonstrating significant potential for expeditious biomedical fabrication [16].

Thermal conduction plays a crucial role in the polymer curing process, especially in highly exothermic FP reactions, where it directly affects the temperature distribution and rate mechanisms of the reaction system and, consequently, the final physical and chemical properties of the polymer material. By establishing mathematical models and using appropriate numerical methods, a detailed analysis of thermal conduction and chemical reactions during the polymerization process can be conducted. In other FP systems, Goldfeder et al. [9] predicted the degree of monomer conversion and front velocity in an adiabatic acrylate FP system based on initial reactant concentration and initial solution temperature, further studying the FP process in non-adiabatic environments. Viner et al. [19] modified this model to accommodate non-polymeric fillers undergoing phase transitions. Frulloni et al. [5] employed an axisymmetric finite difference model with the alternating direction implicit method to model FP in epoxy resins, investigating the effects of their physicochemical properties and boundary conditions on FP evolution. Goli et al. [20] solved transient coupled diffusion–reaction equations based on the Prout–Tompkins model using the finite element method (FEM) to describe FP in DCPD.

Accurately controlling the temperature and cure degree during the front-end polymerization process is crucial for enhancing the physicochemical properties of composite hydrogel materials. Currently, research on FP-synthesized hydrogels primarily focuses on chemical modifications, and related numerical models are still relatively scarce [21]. This study focuses on the acrylamide hydrogel system synthesized through FP. Based on the curing kinetics reaction data of polymerizable deep eutectic monomers (DEM) obtained from Differential Scanning Calorimetry (DSC) tests, this paper employs Malek’s phenomenological model selection method to establish a corresponding autocatalytic reaction model. By integrating an intricate curing kinetics model with the heat conduction equation, the finite element method is employed to accurately predict propagation velocities and temperature distributions within a two-dimensional adiabatic framework. This methodology enables precise alignment with experimental observations, affirming the model’s accuracy in simulating FP dynamics. 

## 2. Methods

### 2.1. Basic Mechanism of FP Communication

Acrylamide-based hydrogels, favored for their reactivity with electrophiles and free radicals, are predominantly synthesized through free radical polymerization [22]. Free radical polymerization kinetics follow three basic steps (Initiation, Propagation, and Termination) [9]:$$\mathrm{Initiation}:\left\{\begin{array}{r} I\overset{k_{d}}{\to }f2R•\\ R•+M\overset{k_{i}}{\to }P_{1}• \end{array}\right.$$
$$\mathrm{Propagation}:\left\{\begin{array}{c} P_{1}•+M\overset{kp}{\to }P_{2}•\\ P_{2}•+M\overset{kp}{\to }P_{3}•\\ \cdots \cdots \\ P_{j}•+M\overset{k_{i}}{\to }P_{j+1}• \end{array}\right.$$
$$\mathrm{Termination}:\left\{P_{n}•+P_{m}•\overset{k_{i}}{\to }P\right.$$

*I* represents the initiator, R• is the initial free radical, *M* is the monomer, $P_{j}•$ is the polymer free radical of length j, and *P* is the chemically inactive polymer. All substances with chemical activity are denoted with a dot. *f* is the efficiency of the initiator, typically valued at 0.5. Free radicals are highly active. Only a small fraction of the formed free radicals decompose and combine with the monomer, being consumed in the process. The reaction rate constant $k_{i}$ depends on the system’s temperature and follows the Arrhenius law. At the reaction’s peak, the above three steps occur simultaneously. Figure 1 illustrates the FP reaction mechanism.

The frontal polymerization process of a DEM is a complex multiphysics process involving physical and chemical changes in materials. Potassium persulfate (KPS) serves as the reaction initiator, transforming the DEM solution mixed with AM and ChCl from a pre-reaction solution into a post-reaction solid-state gel polymer. The rapid transformation and solidification degrees are represented by α. The front-end polymerization process of a DEM is broken down into two parts: the solidification kinetics analysis of the polymerization process and the thermal conduction diffusion model analysis.

### 2.2. Curing Kinetics Model 

The apparent activation energy, frequency factor, and reaction order are the most crucial parameters in curing reaction kinetics. The apparent activation energy reflects the difficulty level of the reaction. The success of the curing reaction depends on whether the energy in the system is higher than the required apparent activation energy. A high-frequency factor accelerates the reaction rate. Many mechanistic reactions coexist during the curing process, making it challenging to ensure a constant apparent activation energy. Determining the activation energy during the reaction process is an important task. Curing kinetics models are divided into phenomenological models and mechanistic models. The phenomenological approach is widely used as it employs new, semi-empirical model equations as a basis when studying curing reaction kinetics, and it obtains the parameters in the model equation through mathematical simulation without involving chemical composition. The curing kinetic rate equation can be expressed as a function related to the temperature and the cure degree:(1)$$\frac{d\alpha }{dt}=\frac{dH/\Delta H}{dt}=K(T)f(\alpha )$$
where d*α*/d*t* is the solidification rate, *t* is the reaction time, *α* is the degree of conversion, and Δ*H* is the total heat of the solidification reaction. We can obtain *α* by calculating the ratio of the instantaneous heat released by the reaction to the total heat. *f*(*α*) is the solidification kinetics model, dependent on the reactants, while *K*(*T*) is a function of temperature and follows the Arrhenius law [23]:(2)$$K(T)=A\exp(-\frac{E_{a}}{RT})$$

In Equation (2), *A* represents the pre-exponential factor, *E_a_* is the apparent activation energy, and *R* is the Avogadro gas constant. By substituting Equation (2) into Equation (1), we can obtain
(3)$$\frac{d\alpha }{dt}=\frac{dT}{dt}\frac{d\alpha }{dT}=\beta (\frac{d\alpha }{dT})=A\exp(-\frac{E_{a}}{RT})f(\alpha )$$

Logarithmizing both sides of Equation (3) yields
(4)$$\ln(\frac{d\alpha }{dt})=\ln\left[\beta (\frac{d\alpha }{\mathrm{dT}})\right]=\ln[Af(\alpha )]-\frac{E_{a}}{RT}$$

Since the DSC experiment cannot directly measure d*α*/d*t*, it is calculated based on the d*α*/dT values, which are easily obtained from DSC experiments. Selecting an appropriate *f*(*α*) is one of the crucial steps when studying FP curing kinetics. Although multiple reactions occur simultaneously during the curing process, for model simplification, it is generally considered that a single reaction can represent the entire curing process. Different types of chemical reactions have different models of description, so it is necessary to choose a reaction model that suits the specific curing system or reaction type.

Malek et al. [24], building on the work of their predecessors, proposed a method to quickly and scientifically infer the most probable mechanism function *f*(*α*). This comprehensive thermoanalytical kinetics method can avoid the hassle of trying each reaction model one by one. The judgment method starts with the determination of *E_a_* values using the isoconversional method, thereby progressively obtaining complete kinetic results. The main steps of Malek’s method are as follows:(1)Use the isoconversional method to determine the activation energy *E_a_*;(2)Determine the form of the reaction kinetics mechanism function *f*(*α*) based on the shape and corresponding characteristic values of the defined functions y(α) and z(α) transformed from experimental data;(3)Based on the kinetic mechanism function, choose the appropriate formula to calculate the kinetic power indices *n*, *m*, etc.;(4)Calculate the pre-exponential factor *A*.

The flowchart for mechanism model selection is shown in Figure 2, and related letter meanings can be referenced from the original literature [24].

### 2.3. Heat Diffusion Transfer Model

The initiation and unidirectional propagation of DEM polymerization assume isotropic thermal conductivity within the solution. Heat transfer in FP reactions is governed by the classical heat conduction equation, necessitating the resolution of a set of nonlinear partial differential equations regarding *α* and temperature [25]:(5)$$\left\{\begin{array}{l} \nabla •(\kappa \nabla T)+\rho H_{r}\frac{\partial \alpha }{\partial t}=\rho C_{p}\frac{\partial T}{\partial t}\\ \frac{\partial \alpha }{\partial t}=A\exp(-\frac{E_{a}}{RT})g(\alpha ) \end{array}\right.$$

In the equation, *κ* (W/(m·K)) denotes thermal conductivity, *ρ* (kg/m³) is density, $C_{p}$ (J/kg·K) is specific heat capacity, and *H_r_* (J/kg) is the total enthalpy of the exothermic reaction. The relationship in Equation (14) outlines the cure kinetics associated with exothermic reactions. Cure kinetics parameters are derived by conducting nonlinear fitting on the cure rate evolution curves extracted from DSC experiments.

In this experiment, the corresponding boundary conditions and initial conditions are as follows:(6)$$\left\{\begin{array}{l} T(x,0)=T_{0}\\ \alpha (x,0)=\alpha_{0}\\ T(0,t)=T_{trig},0\leq t\leq t_{trig}\\ \frac{\partial T}{\partial x}\left|(0,t)=0,t\geq t_{trig}\right. \end{array}\right.$$
where *T_trig_* denotes the time taken to apply a triggering temperature at the boundary end in order to initiate frontal polymerization, *T*_0_ represents the initial temperature of the solution, and *α*_0_ indicates the initial cure degree of the solution [25].

## 3. Experiment and Discussion

### 3.1. Material Preparation

Materials: Acrylamide (AM), Choline Chloride (ChCl), and Potassium Persulfate (KPS) were all purchased from Shanghai Aladdin Biochemical Technology Co., Ltd. (Shanghai, China). All reagents were of analytical grade and were used directly after purchase.

The preparation of the DEM is illustrated in Figure 3. Specifically, ChCl is selected as the hydrogen bond acceptor (HBA) and AM as the hydrogen bond donor (HBD). The mixture of the two materials is agitated vigorously in a constant-temperature oil bath at 75 °C until a uniformly transparent and lucid liquid is formed. Once the temperature cools to around 30 °C, the initiator KPS is added and stirred gently until evenly mixed, and then the heat source application initiates the reaction. The molar ratio of AM to ChCl was 2:1, with KPS constituting 0.5% of the mass. The melting point of the DEM was 30 °C. This study aims to explore the free radical polymerization reaction in a binary mixture system without crosslinking and to measure the heat released during linear polymer chain formation via frontal polymerization.

### 3.2. Non-Isothermal Isothermal DSC Experiment

Non-isothermal DSC experiments are commonly used to study curing reaction kinetics due to their excellent precision as well as sample preparation convenience [26]. To study the curing reaction kinetics of AM-based DEM, multiple DSC pre-tests were conducted using a Differential Scanning Calorimeter (Machine: Mettler Toledo DSC3, METTLER TOLEDO, Greifensee, Switzerland) on the DEM to determine its temperature reaction profile around 30 °C. In the early stages of the curing reaction, endothermic reactions occur, resulting in negative DSC curve values, indicating that the FP reaction has not yet been initiated, as shown in Figure 4. The curing exothermic curves of the DEM at different heating rates *β* (1, 3, 5, 7, 9 K/min) during the experimental exothermic phase were extracted, along with the corresponding characteristic temperature chart, to measure the activation energy as the heat released by the curing reaction. All curves in Figure 4 have only one exothermic peak, indicating that the DEM has a relatively well-distributed curing system [26,27]. It can be observed that as the heating rate increases, the initial reaction temperature ($T_{i}$), the maximum exothermic temperature ($T_{p}$), and the reaction termination temperature ($T_{f}$) all increase accordingly. As the heating rate increases, the residence time of the DEM system at a specific temperature decreases, leading to insufficient curing. The system needs to continue curing at the next temperature. This thermal hysteresis effect causes the temperature difference to become more pronounced; the thermal inertia per unit time increases, causing the exothermic peak to move to higher temperatures [28]. Figure 5 reveals a linear correlation between the curing reaction’s characteristic temperatures and the heating rate across various stages. By extrapolating to a zero heating rate, the onset of the reaction is identified at 307.7 K, the peak exothermic temperature at 325.5 K, and reaction completion at 342.5 K. This is significant in guiding FP reaction initiation at room temperature.

During the curing process, numerous polymerization reactions occur simultaneously, making it difficult to ensure a constant apparent activation energy. The Flynn–Wall–Ozawa (FWO) isoconversional method is utilized in the activation energy calculation [29]. This method allows for activation energy evaluation without prior knowledge of the sample’s reaction mechanism. This is particularly useful for studying new materials or complex systems. By examining the change in apparent activation energy with the conversion rate in a fixed system, it is possible to circumvent the need to select a mechanism function, thus avoiding computational errors due to improper model selection. According to the FWO method, the equation can be represented as follows:(7)$$\left\{\begin{array}{l} T(x,0)=T_{0}\\ \alpha (x,0)=\alpha_{0}\\ T(0,t)=T_{trig},0\leq t\leq t_{trig}\\ \frac{\partial T}{\partial x}\left|(0,t)=0,t\geq t_{trig}\right. \end{array}\right.$$
where g(*α*) is the integral function of the conversion rate. When selecting the same conversion rate *α* under different heating rates, g(*α*) can be considered a constant. *E_α_* can be calculated from the slope of the ln*β* versus 1/T curve. We obtained the ln(*β*) − 1/*T* curve for conversion rates ranging from 0.2 to 0.8 using the FWO method and determined the activation energy *E_a_* at various conversion rates. The calculation results are depicted in Figure 6. Overall, we derived an average activation energy of 89.567 kJ/mol from the curve analysis.

Figure 7′s analysis of DSC data reveals the relationship between ln[A*f*(*α*)] and ln(1 − *α*). The curve’s pattern indicates a nonlinear correlation between these variables, suggesting the reaction system deviates from the conventional nth-order reaction model prevalent in model-fitting techniques. This observation underscores the reaction’s autocatalytic nature [30].

To solve the dynamic parameters and determine the specific dynamic model, the Malek inference method creates two important parameters y(α) and z(α).
(8)$$y(\alpha )=\beta \frac{d\alpha }{dT}\exp(x)=\frac{d\alpha }{dt}e^{x}$$
(9)$$z(\alpha )=\pi (x)(\frac{d\alpha }{dt})\frac{T}{\beta }$$

In Formulas (8) and (9), $x=E_{a}/RT$, where the average value of *E_a_*, obtained earlier via the FWO method, is 89.567 kJ/mol. *π*(*x*) can be represented by Equation (10), proposed by Senum–Yang [23].
(10)$$\pi (x)=\frac{x^{3}+18x^{2}+88x+96}{x^{4}+20x^{3}+120x^{2}+240x+120}$$

Figure 8 presents the normalized plots of y(α) and z(α), with characteristic values obtained from the experiment, as shown in Table 1. *α*_p_, *α*_m_, and *α*_p_^∞^ correspond to the maximum values of d*α*/d*t*, *y*(*α*), and *z*(*α*) with respect to *α*, respectively.

The data reveal that the maximum characteristic values of *y*(*α*) and *z*(*α*) satisfy 0 < *α*_m_ < *α*_p_ and *α*_p_^∞^ ≠ 0.632. According to the criteria set by Malek’s method, the curing process of this system can be represented by the Sesták–Berggren bimolecular autocatalytic kinetics model [31]. According to the Sesták–Berggren autocatalytic model, if the maximum peak of the reaction occurs between 30 and 40% of the process, the curing reaction follows:(11)$$f(\alpha )={\left(1-\alpha \right)}^{n}\alpha^{m}$$
where *m* and *n* are the independent reaction orders of the autocatalytic reaction model. At this point, the kinetic equation can be transformed into
(12)$$\frac{d\alpha }{dt}=A\;\exp(-\frac{E_{a}}{RT}){\left(1-\alpha \right)}^{n}\alpha^{m}$$

Taking the logarithm of both sides of the equation yields
(13)$$\ln(\frac{d\alpha }{dt})=\ln A-\frac{E_{a}}{RT}+n\ln(1-\alpha )+m\ln\alpha$$

By setting *s* = *m*/*n*, we can obtain
(14)$$\ln[\frac{d\alpha }{dt}\exp(\frac{E_{a}}{RT})]=\ln A+n\ln\alpha^{s}(1-\alpha )$$

The slope can be obtained by performing a linear fit of the relationship curve between ln[(d*α/*d*t*)e*^x^*] and ln[*α*^s^(1 − *α*)] in the interval (0.2, 0.8), and it can be used to determine the value of *n*, where $\alpha_{m}=m/(m+n)$. A substitution calculation yields $s=\alpha_{m}/(1-\alpha_{m})$. After determining *n* from the fitted curve, *m* can be calculated using *m* = *sn*, and the value of *A* can be obtained from the intercept. Figure 9 illustrates the relationship curve between ln[(d*α/*d*t*)e*^x^*] and ln[*α*^s^(1 − *α*)] for the degree of cure *α* within the interval (0.2, 0.8).

Based on the intercept and slope fitted from the graph, the kinetic parameters *m*, *n*, and lnA were calculated for the autocatalytic model. The results are displayed in Table 1.

We compared the experimental DSC results at different heating rates with the predicted outcomes from the equation-fitted rate curves to verify the accuracy of the derived kinetic Equation (15) for the curing reaction. By substituting the values of *n*, *m*, and A, obtained at different heating rates, into Equation (12), we can calculate the fitted values of d*α*/d*t* for comparison with experimental values, as shown in Figure 10. The comparison of the fitted curves reveals that the autocatalytic model’s predictions generally align with the experimental curves across various heating rates. However, certain deviations are observed in the later stages of the curing reaction. At higher temperatures, the experimental values tend to diverge from the theoretical ones due to a significant decrease in reactant concentration, necessitating additional energy from external sources to facilitate effective collisions between active groups and continue the curing reaction. This shift indicates a gradual transition from chemically controlled kinetics to diffusion-controlled processes.

Overall, the non-isothermal curing reaction of the AM-based DEM system can be described using an autocatalytic reaction model. The corresponding autocatalytic kinetics model is
(15)$$\frac{d\alpha }{dt}=5.86\times 10^{13}\exp(-\frac{89567}{RT}){\left(1-\alpha \right)}^{1.234}\alpha^{0.365}$$

### 3.3. Numerical Simulation

We use the finite element method to solve the nonlinear partial differential equation obtained from Equation (5) and implement it using the finite element software ABAQUS (version:2021). This software is effective for general heat transfer problems that do not involve intricate internal heat generation. Figure 11b showcases typical temperature and cure degree profiles within a liquid DEM domain (initial temperature T_0_ = 305 K) over a length of r = 5 mm, adequate for capturing the stable propagation of the polymerization front. A minimal initial cure degree of *α*_0_ = 0.0001 ensured that the cure degree field calculations were non-zero. The experimental setup, conducted in cylindrical tubes, was represented computationally through an axisymmetric, two-dimensional model. The thermodynamic properties of the AM-based DEM solution are summarized in Table 2.

To initiate the aggregation process, after preliminary testing, a temperature trigger of 474 K was applied continuously for 0.2 s to the left edge of the domain, which was sufficient to initiate FP. The initial number of mesh cells and the initial time step were set to 15,600 and 0.001 s, respectively. A refined mesh size of 0.1 mm × 0.1 mm was used to discretize the DEM domain in order to capture the narrow advancing front. Based on the conditions simulated at room temperature, the numerical simulation yielded an average peak front advancement velocity of 2.7 mm/s, with specific temperature distribution and cure simulation degree situations, as shown in (c) and (d).

Figure 12 illustrates the impact of the trigger temperature application duration on the propagation speed of the front end. When the duration of the trigger increases from 0.1 s to 1 s, the average front-end speed increases from 2 mm/s to 3.2 mm/s. Beyond 1 s, the acceleration reaches saturation as the heat generated by the material is sufficient to activate the front-end polymerization process. At short trigger durations, only the material’s surface is heated, while deeper areas remain at lower temperatures.

Figure 13 demonstrates the initiator temperature’s impact on frontal velocity, revealing a modest temperature-dependent increase under a fixed 0.2 s trigger duration. This velocity includes an initial phase of rapid heating and a subsequent self-sustaining FP polymerization phase. The results indicate the convergence of these phases at elevated temperatures, with the self-propagation speed reaching a peak rather than continuously increasing. The influence of the initial applied temperature diminishes as propagation is primarily driven by the polymerization’s exothermic heat, leading to velocity stabilization.

Figure 14 underscores the significant effect of the system’s initial temperature, T_0_, on FP velocity. Higher initial temperatures markedly accelerate the propagation speed; for instance, at T_0_ = 321 K, the speed is approximately 1.5 times greater than at 289 K, suggesting that bulk polymerization dominates above the DEM’s FP initiation temperature. This implies that a higher T_0_, without surpassing the initiation threshold, effectively enhances FP polymerization speed, aligning with SCI journal requirements for concision and academic rigor.

### 3.4. Model Validation

We initiate the FP reaction by thermally triggering acrylamide-based DEM to validate the numerical simulation results obtained from the fitted curing kinetics equations and analyze their predictions in terms of temperature distribution and propagation speed. As shown in Figure 15, the mold used in the experiment consists of several insulated aluminum silicate ceramic fiber boards, and the test tube is clamped from both sides using insulation boards. These fiber boards provide excellent insulation around the test tube, reducing thermal losses in the diffusion environment of the FP reaction. After preparing the DEM reagent with the initiator KPS, we rapidly transferred the solution into the test tube using a volumetric syringe. When the liquid level in the test tube stabilized, an electric soldering iron was used to heat the upper end of the test tube to trigger the FP reaction. The characteristic temperature distribution and diffusion speed during the FP reaction were captured in real-time using an infrared thermal imaging camera with a temperature measurement range from −50 °C to 400 °C.

Figure 16 presents the experimental measurements of T_max_ and its temperature distribution, noting a thermal peak at 141.1 °C and a propagation velocity of 1.5 mm/s. The observed distribution of thermal hotspots from infrared imaging aligns closely with the patterns predicted by numerical simulations, validating the model’s capacity to precisely represent the temperature profiles and propagation speeds characteristic of FP reactions. Nevertheless, the simulations are observed to predict higher T_max_ values, a discrepancy attributed to the model’s limitations in capturing microscale phenomena and accounting for thermal diffusion through experimental insulation. This discrepancy underscores the necessity to enhance the granularity and accuracy of simulations to better reflect the nuanced behaviors observed in FP reactions.

## 4. Conclusions


(1)A kinetic model of the curing process for acrylamide-based deep eutectic solvents was successfully developed using Differential Scanning Calorimetry under non-isothermal conditions. The model was validated through a comparison with empirical data, demonstrating that the nth-order autocatalytic model exhibits reliability and accuracy in curing kinetics for DEM-based synthesis. This finding underscores the model’s potential applicability in the predictive analysis of polymerization kinetics within similar systems.(2)Finite element numerical simulations were employed to elucidate factors influencing the characteristic temperature and velocity of the frontal polymerization (FP) process, examining the regulatory role of temperature within the FP reaction. The results indicate that both the duration of and increase in the triggering temperature momentarily elevate the front velocity before it stabilizes, while the initial temperature of the liquid substantially affects the front velocity. This observation provides valuable insights into the thermal dynamics governing FP and informs reaction condition optimization.(3)When comparing numerical simulation predictions with experimental outcomes, discrepancies predominantly attributed to experimental measurement errors, as well as the susceptibility of experiments to temperature control, were noted, resulting in higher predictive values overall. However, the overall temperature distribution and peak temperature characteristics exhibited remarkably similar trends between simulations and experiments. This congruence affirms the accuracy of numerical simulations and their reliability in forecasting the rapid curing process of hydrogel composites synthesized via FP. Consequently, these simulations serve as an essential reference for experimental design and optimization, facilitating advancements in the efficient fabrication of high-performance hydrogel composites. This indicates that under the strict control of experimental conditions, there is still some space to optimize the model in order to improve its accuracy.


## Figures and Tables

**Figure 1 polymers-16-00873-f001:**
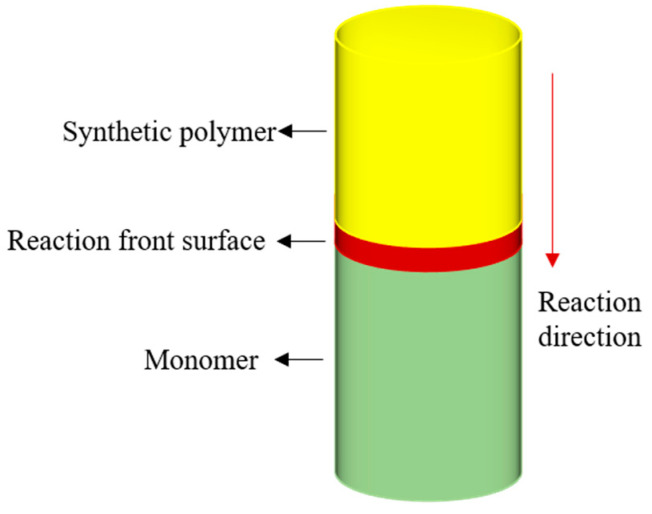
Schematic diagram of the FP reaction.

**Figure 2 polymers-16-00873-f002:**
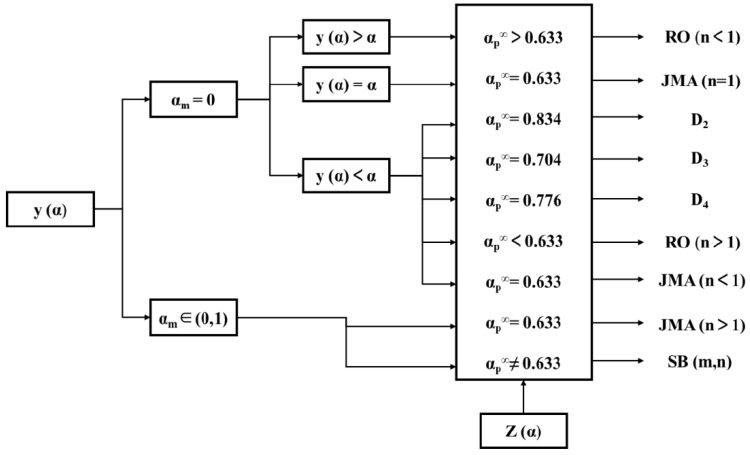
Process diagram of Malek method screening mechanism model.

**Figure 3 polymers-16-00873-f003:**
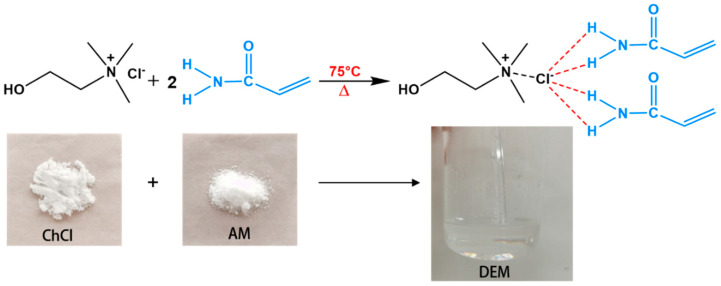
Schematic diagram of the synthesis and preparation of DEM.

**Figure 4 polymers-16-00873-f004:**
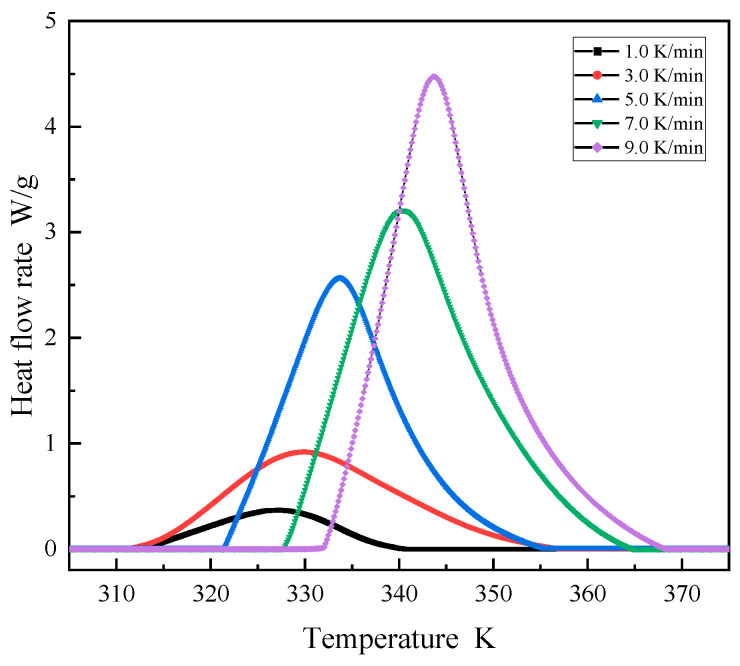
DSC curves of DEM under different heating rates.

**Figure 5 polymers-16-00873-f005:**
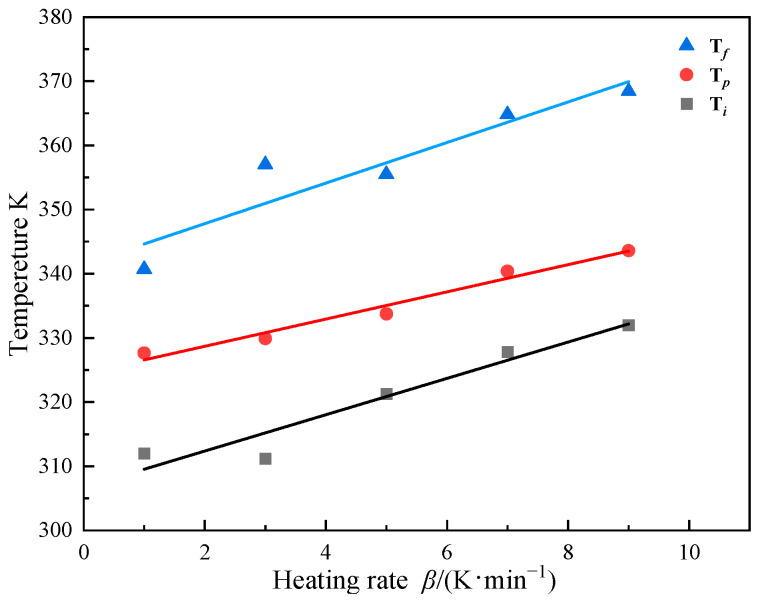
The relationship between characteristic temperature T and heating rate *β*.

**Figure 6 polymers-16-00873-f006:**
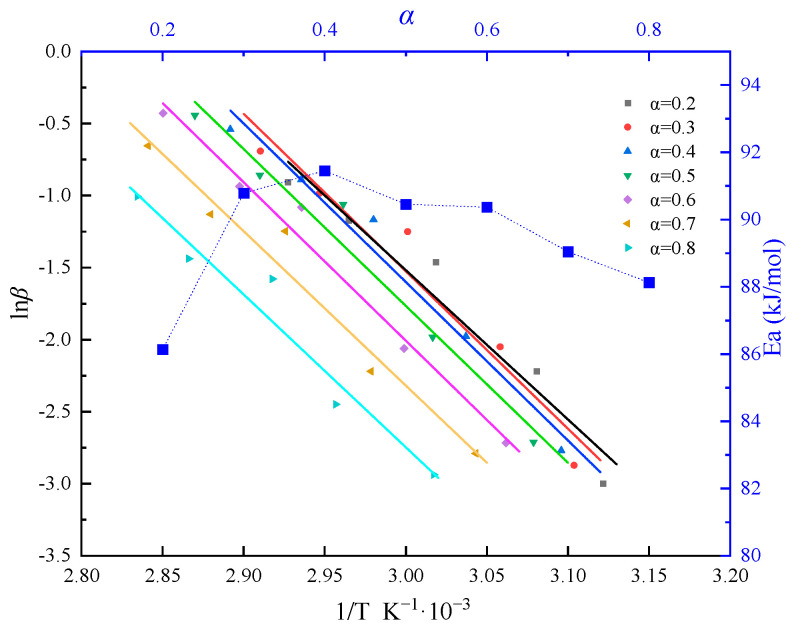
Analysis of activation energy curve using FWO method.

**Figure 7 polymers-16-00873-f007:**
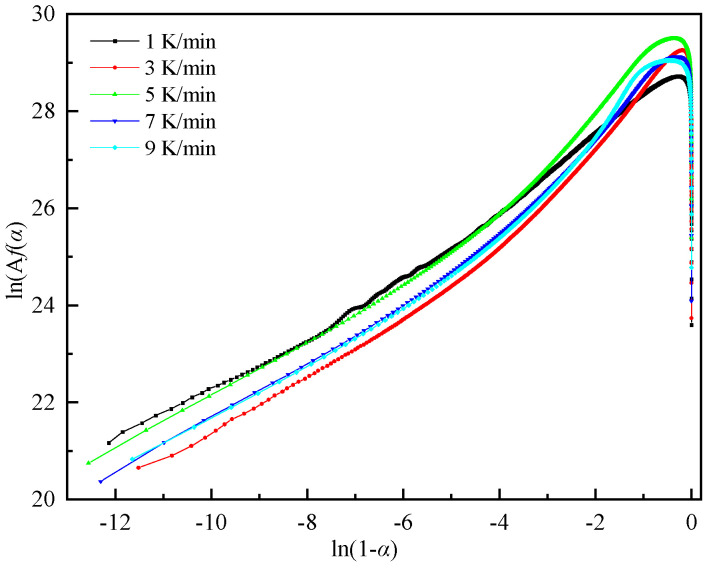
Plots of ln[A*f*(α)] versus ln(1 − α) of the DEM at different heating rates.

**Figure 8 polymers-16-00873-f008:**
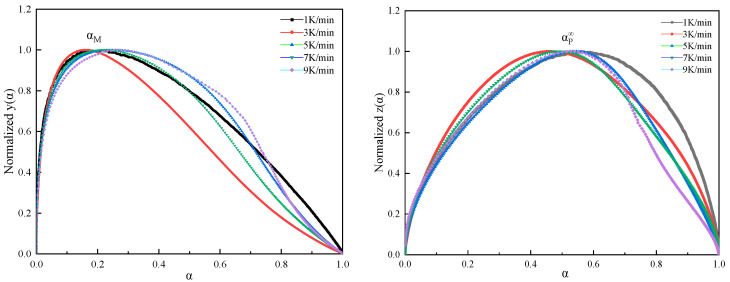
Plots of normalized *y*(*α*) and z(*α*) against *α*.

**Figure 9 polymers-16-00873-f009:**
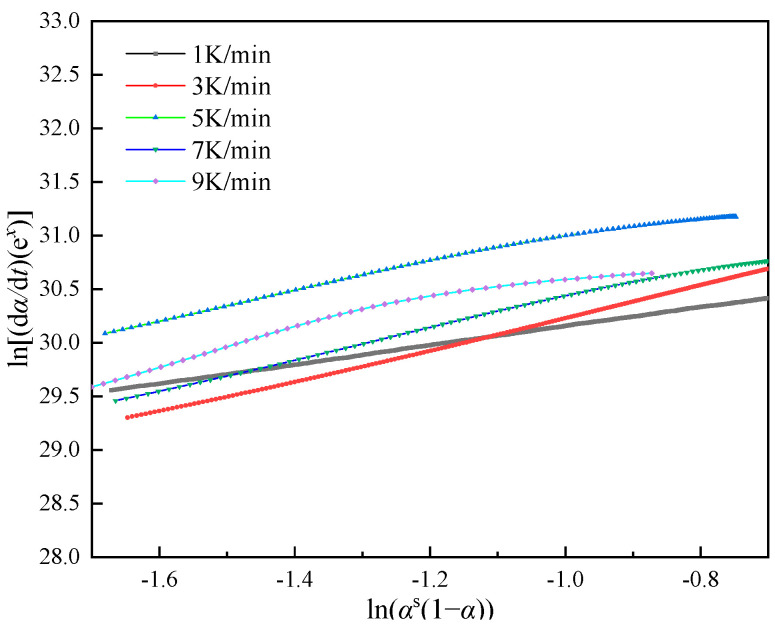
Plots of ln[(d*α*/d*t*)e*^x^*] against [*α*^s^(1 − *α*)].

**Figure 10 polymers-16-00873-f010:**
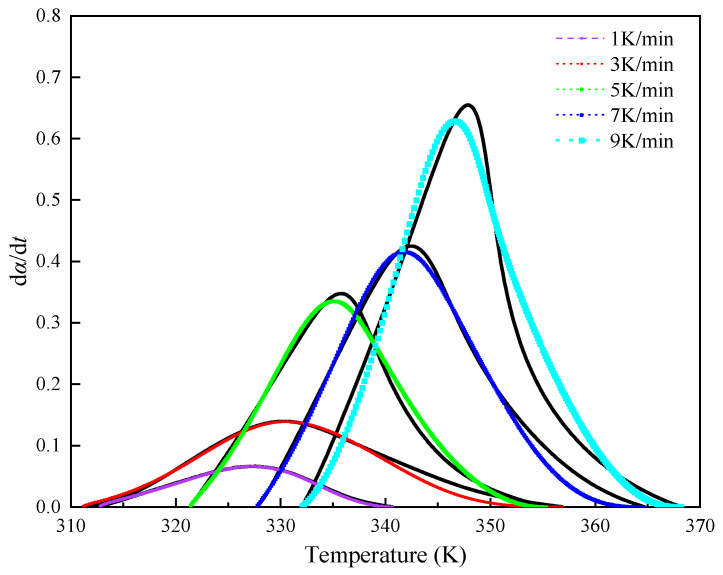
Comparing the autocatalytic reaction rate model with DSC data: black lines denote experimental values; colored lines indicate fits.

**Figure 11 polymers-16-00873-f011:**
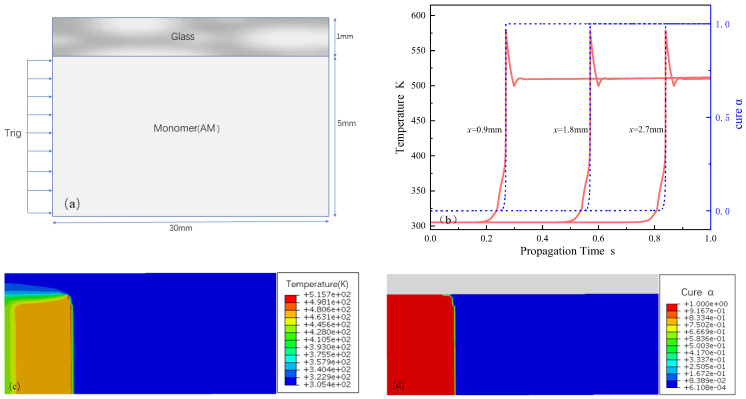
Numerical simulation calculation chart for FP based on AM. (**a**) Simulation calculation model diagram; (**b**) curing degree and temperature curve of the propagation front-end; (**c**) simulation distribution of propagation temperature; (**d**) simulation distribution of propagation curing degree.

**Figure 12 polymers-16-00873-f012:**
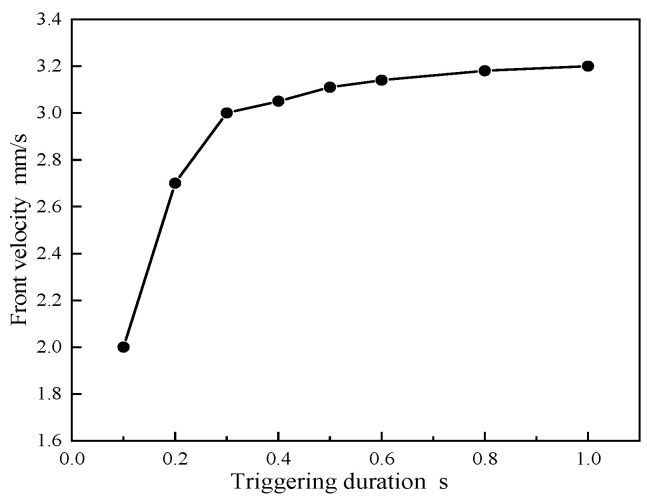
Impact of trigger time on front-end speed.

**Figure 13 polymers-16-00873-f013:**
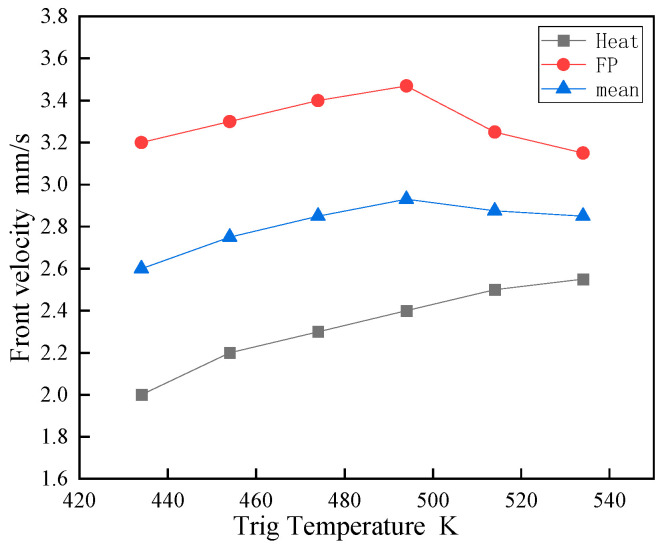
The influence of different triggering temperatures on front-end speed.

**Figure 14 polymers-16-00873-f014:**
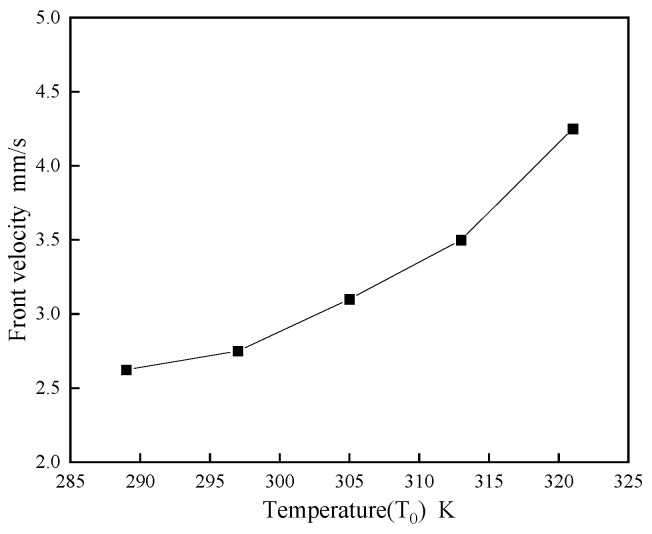
The influence of system temperature T_0_ on front-end speed.

**Figure 15 polymers-16-00873-f015:**
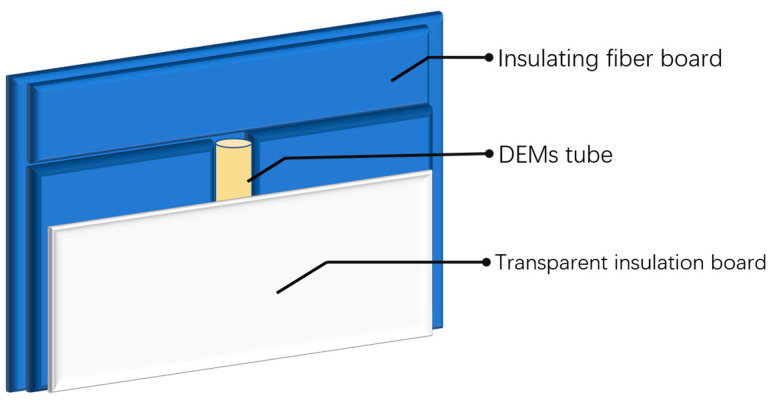
Schematic diagram of FP experimental model.

**Figure 16 polymers-16-00873-f016:**
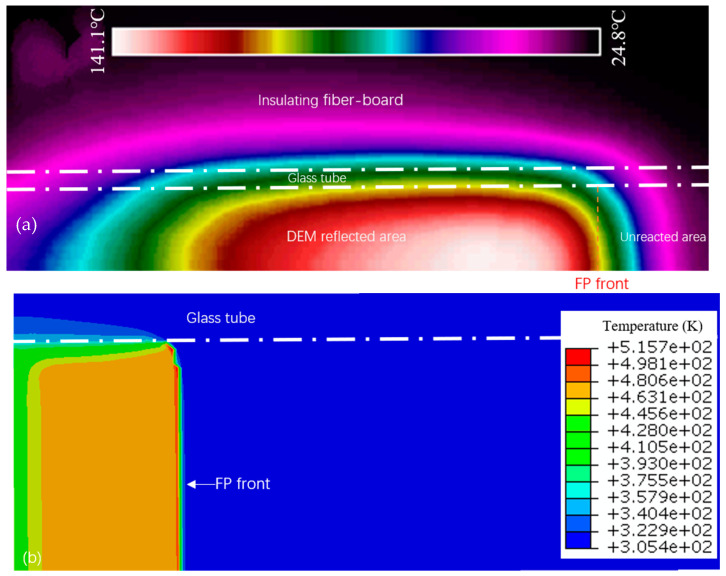
Comparison of simulation and experimental data: (**a**) experimental measurement of FP infrared thermal image; (**b**) numerical simulation results for FP.

**Table 1 polymers-16-00873-t001:** Characteristic kinetic parameters from autocatalytic models across varied heating rates.

β	α_p_	α_m_	α_p_^∞^	n	m	lnA
1	0.561	0.214	0.561	0.883	0.240	31.035
3	0.478	0.189	0.468	1.490	0.347	31.725
5	0.534	0.249	0.540	1.133	0.376	32.073
7	0.485	0.222	0.489	1.357	0.386	31.750
9	0.556	0.267	0.527	1.309	0.476	31.927
mean	0.523	0.223	0.517	1.234	0.365	31.702

**Table 2 polymers-16-00873-t002:** Thermophysical parameters of DEM based on AM.

Monomer	κ (W/m·k)	*ρ* (kg/m^3^)	$C_{p}$	H*_r_* (J/g)	T_0_ (K)
AM	0.25	900	1990	406.05	305

## Data Availability

Data are contained within the article.

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
