# Peer review of "Numerical Simulation of Polyacrylamide Hydrogel Prepared via Thermally Initiated Frontal Polymerization"

_polymers, 2024, doi:10.3390/polym16070873_

Round 1
Reviewer 1 Report
Comments and Suggestions for Authors
The manuscript titled “Numerical Simulation of Polyacrylamide Hydrogel Prepared by Thermal Initiated Frontal Polymerization” uses a finite element method to describe the initiation and propagation of the exothermic front in a frontal polymerization of acrylamide. This study is important since thermoset polymer-based composite materials are widely used in various engineering applications. Most importantly, the study aims at filling up gaps in the lack of effective predictive models for the front conversion speed and temperature propagation speed during the frontal polymerization synthesis process of hydrogels.
Although interesting and important, the manuscript lacks formality and more comparison with experimental results to be considered for publication.
Major issues:
- The modeling approach needs better justification; stating the model is ‘simple’ and it is the prefer approach without proper citations or justification is not enough.
- I would have preferred more discussion regarding why the particular forms for the reaction model were chosen. I am not implying they are wrong choice, but the manuscript does not discuss why these types of equations were chosen over other options for the system under study.
- To evaluate feasibility a brief explanation of the FWO method is needed.
- Figure 8 shows good agreement with data, however the way in which this was obtained is very convoluted. Perhaps the manuscript needs so reorganization, in order to make this part clearer.
- Since the front advances as a shock wave, a numerical stability analysis of the methods should be included to proof that the results are not affected by numerical instabilities.
- I do not see in the simulated data the agreement with Figure 14. I believe that the message will be better delivered if there is a side-by-side comparison. Specifically, the sentence below is not well justified by the data presented in the manuscript.
“The observed distribution of thermal hotspots from the infrared imaging aligns closely with the patterns predicted by numerical simulations, validating the model's capacity to precisely represent the temperature profiles and propagation speeds characteristic of FP reactions.”
Minor issues:
- There are several typos in the manuscript.
- Below Eqn. 13 it will be good to point to figure 7 as a reference of how those curves look. It is hard to assess how 3 parameters can be determined out of a single curve. It seems the fitting procedure is under-specified.
- I am not sure what is the intended meaning of the following sentence “This expression has been refined for enhanced precision, conciseness, and academic rigor.”
Comments on the Quality of English LanguageThere are several typos in the text
Author Response
Thank you very much for the reviewers' comments concerning our manuscript .(ID: polymers-2898618). Those comments are all valuable and very helpful for revising and improving our paper, as well as the important guiding significance to our researches. We have studied comments carefully ang have made correction which we hope meet with approval. The main corrections in the paper and the responds to the reviewer' s comments are in the word file.

Reviewer 2 Report
Comments and Suggestions for Authors
My comments are provided in the attached pdf.

Author Response

(The authors gave the same response as above.)

Reviewer 3 Report
Comments and Suggestions for Authors
The authors need to improve the paper organization and presentation including but are not limited to
i. The abstract, introduction and conclusion sections need to be improved.
ii. Need to adjust the size of Figure 1.
iii. Need to align Fig 2 and Fig 3, Fig 4 and Fig 5
iv. The are some unexpected messages (Citation: Yi, X.; Li, S.-f.; Wen, P.; Yan, S.-l.) right above Equ (1).
Comments on the Quality of English Language
Should ask for help on writing the paper.
Author Response

(The authors gave the same response as above.)

Round 2
Reviewer 1 Report
Comments and Suggestions for Authors
The previous comments have all been addressed appropriately.